# Risk of metabolic disorders in childless men: a population-based cohort study

Ane Berger Bungum,[1,2] Clara Helene Glazer,[1] Jens Peter Bonde,[1] Peter M Nilsson,[3] Aleksander Giwercman,[2] Sandra Søgaard Tøttenborg[1]

[1]Department of Occupational and Environmental Medicine, Bispebjerg University Hospital, Kobenhavn, Denmark
[2]Molecular Reproductive Medicine, Department of Translational Medicine, Lund University, Malmö, Sweden
[3]Department of Clinical Sciences Malmö, Lund University, Skåne University Hospital, Malmö, Sweden

**Correspondence to**
Ane Berger Bungum;
ane.berger.bungum@regionh.dk

## ABSTRACT

**Objective** To study whether male childlessness is associated with an increased risk of metabolic disorders such as metabolic syndrome (MetS) and diabetes.

**Design** A population-based cohort study.

**Setting** Not applicable.

**Participants** 2572 men from the population-based Malmö Diet and Cancer Cardiovascular Cohort.

**Interventions** None.

**Main outcome measures** From cross-sectional analyses, main outcome measures were ORs and 95% CIs for MetS and diabetes among childless men. In prospective analyses, HRs and 95% CI for diabetes among childless men.

**Results** At baseline, in men with a mean age of 57 years, the prevalence of MetS was 26% and 22% among childless men and fathers, respectively. Similarly, we observed a higher prevalence of diabetes of 11% among childless men compared with 5% among fathers. In the cross-sectional adjusted analyses, childless men had a higher risk of MetS and diabetes, with ORs of 1.22 (95% CI 0.87 to 1.72) and 2.12 (95% CI 1.34 to 3.36) compared with fathers. In the prospective analysis, during a mean follow-up of 18.3 years, we did not see any increase in diabetes risk among childless men (HR 1.02 (0.76 to 1.37)).

**Conclusion** This study provides evidence of an association between male childlessness and a higher risk of MetS and diabetes. However, as these associations were found in cross-sectional analyses, reverse causation cannot be excluded.

## INTRODUCTION

A man's reproductive health may not only reflect his chance to become a father, but may also be related to his general health.[1] In recent years, male childlessness and infertility have been reported to be associated with an increased risk of all-cause mortality and cardiovascular disease.[2–5] Male infertility has also been associated with a higher risk of metabolic disorders.[3 6 7] Metabolic syndrome (MetS) and type 2 diabetes are metabolic disorders, with increasing prevalence worldwide.[8 9] MetS is a syndrome consisting of a cluster of markers, including visceral obesity, hypertension and hyperglycaemia.[8] Importantly, the syndrome may help to identify

individuals at future risk of type 2 diabetes and cardiovascular disease.[10–12]

Cross-sectional studies have demonstrated an association between factors related to male reproductive health (eg, hypogonadism, reduced semen quality and erectile dysfunction) and MetS, as well as type 2 diabetes,[13–18] but whether poor reproductive health precedes MetS and diabetes or vice versa is uncertain due to the cross-sectional design of these studies. Two recent prospective studies found increased risk of developing diabetes among infertile men.[3 7] Authors of these prospective studies did not suggest a causal relationship for the association, but rather common aetiologies of infertility and diabetes such as shared genetics and factors related to endocrine regulation, lifestyle or in-utero exposures. However, these prospective studies failed to adjust for body mass index (BMI), physical activity level and other lifestyle factors considered as common risk factors for type 2 diabetes and poor male reproductive health.[11 19] Also, as type 2 diabetes can remain asymptomatic and undiagnosed for years, the measure of outcome of both prospective studies have a central

limitation in common; dependence on health-seeking behaviour of the study participants.

Assessment of fertility status of a man is based on access to clinical and laboratory data, including semen analysis, which are difficult and costly to obtain in population-based studies. However, information regarding childlessness is more easily accessible and may be a feasible proxy for infertility. Therefore, using data from the Malmö Diet and Cancer Cardiovascular Cohort (MDC-CC), we aimed to examine whether male childlessness is associated with MetS and diabetes, while taking potential confounding lifestyle factors into account. We first examined the prevalence of MetS and diabetes in men with and without children, and next we assessed the incidence of diabetes during a mean follow-up of 18.3 years. Study participants were examined for diabetes both at baseline and follow-up clinical examination, limiting the influence of health-seeking behaviour and giving reliable estimates of diabetes prevalence and incidence among childless Swedish men.

## MATERIALS AND METHODS
### Study population
The Malmö Diet and Cancer Cohort (MDC) is a population-based cohort of 30 446 Malmö residents (12 120 men born between 1926 and 1945 and 18 326 women born between 1923 and 1950) enrolled during 1991-1996.[20] The participation rate was 38% for men.[21] Following the initial acceptance letter, during the years of 1991–1994, half of the individuals in MDC randomly selected were invited to participate in a subcohort named MDC-CC. Of these, 2572 men accepted the invitation (figure 1).

Both at baseline (1991–1994) and follow-up (2007–2012), MDC-CC participants completed a questionnaire regarding marital status, number of children and lifestyle factors (alcohol (g/day), smoking habits (regularly, occasionally, stopped, never) and total physical activity score (according to the Minnesota Leisure Time Physical Activity Questionnaire[22]—calculated as minutes/week for spring/summer/autumn/winter multiplied with an activity-specific factor according to the type of activity, eg, running, walking). Participants also underwent a clinical examination including body composition, blood pressure measurement and collection of venous blood samples.[21] At follow-up examination where 1522 men (59%) participated, the clinical examination also included an oral glucose tolerance test in study participants without known diabetes.

### Patient and public involvement
Patients and or public were not involved.

### Information on childlessness
Information regarding childlessness came from two sources: the baseline questionnaire and the Swedish Tax Agency (STA). In the questionnaire, participants were asked *Do you have any children?'* with reply options 'Yes' or 'No'. The STA holds the number of registered children and their respective birth dates. Data were linked using the unique 10-digit personal identification number assigned to all Swedish citizens. We linked these data sources to stratify the participants into four groups: 'Childless', 'One or more child', 'Conflicting information' and 'Unknown'. 'Conflicting information' appeared if a participant answered 'No' to *Do you have any children?'* in the baseline questionnaire, but was registered with

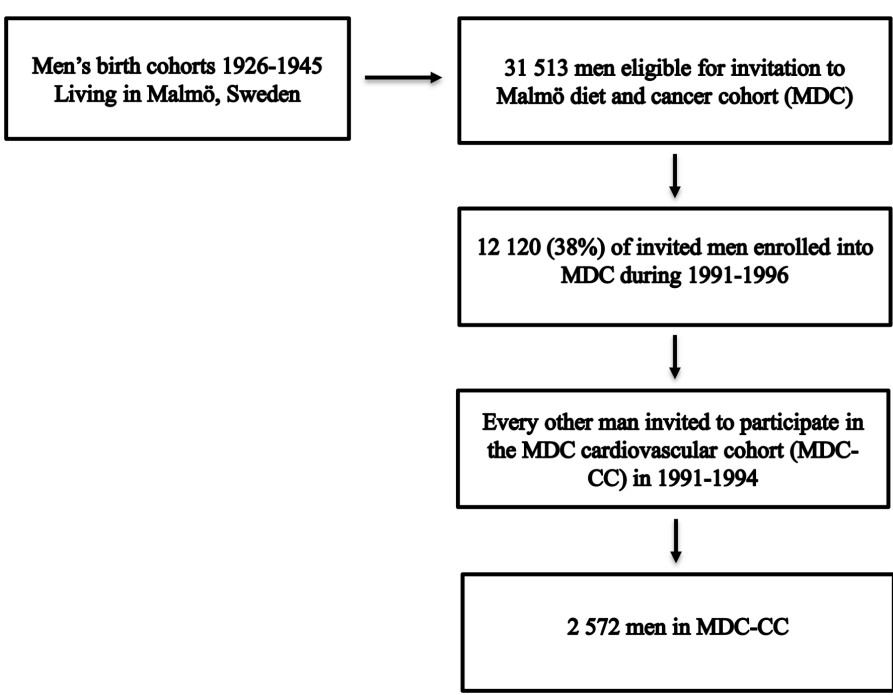

**Figure 1** Malmö Diet and Cancer Cardiovascular Cohort (MDC-CC) recruitment.

one or more children in the STA. Men with 'Conflicting information' who became fathers after the entry date into the MDC-CC cohort were treated as 'Childless' as they were childless at baseline. The remaining men with 'Conflicting information' were registered as fathers in the registry of STA and were therefore treated as having 'One or more child'. We used the designation 'Unknown' if no information regarding children was available from the STA and if no answer was provided to *'Do you have any children?'* in the baseline questionnaire.

## Assessment of MetS

MetS was defined according to the harmonised criteria of the International Diabetes Federation Task Force on Epidemiology and Prevention; National Heart, Lung, and Blood Institute; American Heart Association; World Heart Federation; International Atherosclerosis Society; and International Association for the Study of Obesity.[23] Accordingly, MetS was present if three or more of the following criteria were met: fasting blood glucose (fB-glucose) level above 5.6 mmol/L or the use of anti-diabetic drugs, high-density lipoprotein cholesterol level below 1.03 mmol/L or the use of lipid-modifying treatment, triglycerides level above 1.7 mmol/L or the use of lipid-lowering drugs, waist circumference higher than 102 cm and blood pressure above 130/85 mm Hg or the use of antihypertensive medicine.[23] Data regarding MetS criteria were only available from the baseline clinical examination.

## Diagnosis of diabetes

The diabetes cases and the date of diagnosis were identified from 14 different data sources, including the Swedish National Diabetes Register (NDR), The Swedish Prescribed Drug Register, and from baseline and follow-up screenings in MDC, MDC-CC and the Malmö Preventive Project (MPP).[24] These sources were used to identify prevalent cases of diabetes (type 1 and type 2) at baseline and new-onset incident cases of diabetes (type 1 and type 2) during the follow-up period. In brief, individuals with a date of diagnosis registered in the NDR and/or Diabetes 2000 Register were considered to have diabetes. The same was true for individuals in the local haemoglobin A1c (HbA$_{1c}$) Register in Malmö with at least two HbA$_{1c}$ ≥6%, those with the Tenth Edition of International Classification of Diseases (ICD10) codes E10-E14 and O244-O249, and corresponding ICD7-9 codes in the National Hospitalisation Register or in the Cause-of-Death Register, and men in The Swedish Prescribed Drug Register with Anatomical Therapeutic Chemical code A10.

From baseline questionnaires of MDC and MPP, participants who answered 'Yes' to *'Do you have diabetes?'* and/or listed antidiabetic drugs were considered as patients with diabetes. At the baseline examination of MDC-CC, individuals were considered having diabetes if the fB-glucose measurement was ≥6.5 mmol/L. In MPP and at the MDC-CC follow-up, fB-glucose ≥6.5 mmol/L had

to be verified through oral glucose tolerance test and/or fasting plasma glucose measurements. A full list of the diabetes diagnostic sources is provided in online supplementary table I. The participants were considered as having diabetes from their first diagnosis of diabetes while any and subsequent contradictory information about diabetes was ignored.[25]

## Statistical analyses

### Cross-sectional analysis

To assess the association between male childlessness and metabolic disorders, the prevalence of MetS and diabetes at baseline 1991–1996 was compared among men with and without children by means of logistic regression and reported as ORs with corresponding 95% CIs. As childlessness could be a function of not having a partner, sensitivity analyses including only married men were also performed.

### Prospective analysis

To assess whether the possible association between male childlessness and diabetes persisted or attenuated later in life, HR with 95% CIs of diabetes among childless men compared with fathers were computed using Cox proportional hazard models. Men with pre-existing diabetes at baseline were excluded. The men were followed from enrolment into the MDC-CC until date of diabetes diagnosis, emigration, death or end of follow-up (31 December 2014). Kaplan-Meier plots allowed for visual evaluation of proportional hazards assumption. As with the cross-sectional analysis, we completed a sensitivity analysis, including only married men.

Potential confounders were identified a priori using directed acyclic graphs.[26] Adjustments were performed in two steps in all analyses. The first adjustment step (model I) included age in years, marital status (married/unmarried/divorced/widower), socioeconomic index (workers, unskilled/workers, skilled/lower positioned official or salaried/intermediate positioned official or salaried/employers or self-employed) and educational attainment (no education/primary school/secondary school/high school/>1 year education after high school/university degree). The second adjustment step (model II) also included BMI in kg/m$^2$, alcohol consumption in grams per day, smoking habits (regularly smoker/occasional smoker/stopped smoking/never smoked) and physical activity level in minutes per week. The reason for the distinction between model I and model II is that it can be argued that BMI, alcohol consumption, smoking habits and physical activity level mediates rather than confound the association between childlessness, MetS and diabetes. For instance, smoking habits can affect a man's reproductive health, but on the contrary if a man becomes a father, this can affect his smoking habits.

In the statistical analyses, probability values (p values) were not included, instead 95% CI for measures of association were reported to display the measure of precision.[27] All statistical tests were performed using SAS V.9.4.

**Table 1** Sociodemographic and lifestyle characteristics of men with and without children at baseline

| | Childless men (n=333) | Fathers (n=1817) |
|---|---|---|
| Age (years) (n=2 150) | 57.3 (6.1) | 57.1 (5.9) |
| Marital status  (n=2052) | | |
| Married (%) | 51 | 81 |
| Unmarried (%) | 8 | 3 |
| Divorced (%) | 4 | 14 |
| Widower (%) | 37 | 2 |
| Socioeconomic index  (n=1927) | | |
| Employers and self-employed (%) | 16 | 22 |
| Official/salaried, intermediate position (%) | 21 | 24 |
| Official/salaried, lower position (%) | 21 | 15 |
| Workers, skilled (%) | 17 | 21 |
| Workers, unskilled (%) | 25 | 18 |
| Highest level of education  (n=2056) | | |
| No education (%) | 0 | 1 |
| Primary school (%) | 50 | 45 |
| Secondary school (%) | 22 | 20 |
| High school (%) | 11 | 12 |
| >1 year education after high school (%) | 7 | 10 |
| University degree (%) | 10 | 12 |
| Body Mass Index (kg/m$^2$) (n=2148) | 26 (4.1) | 26.1 (3.3) |
| Alcohol (g/day) (n=2057) | 14.6 (18.4) | 15.8 (15.2) |
| Present smoker (n=2058) | | |
| Regularly (%) | 24 | 22 |
| Occasionally (%) | 4 | 5 |
| Stopped (%) | 38 | 43 |
| Never (%) | 34 | 30 |
| Physical activity score* (n=2 041) | 8675.3 (6771.0) | 8361.9 (6224.9) |

Means (SD) and proportions.
*Min/week for spring/summer/autumn/winter multiplied with an activity-specific factor according to the type of activity, for example, running, walking.

## RESULTS

Among all 2572 men, 422 had missing information regarding fatherhood status and were excluded from analyses. Twenty men had 'Conflicting information' regarding fatherhood, of which 18 men had registered children in the STA before baseline, and two men became fathers after baseline and thus treated as 'Childless'. Consequently, 2150 men were included in the analyses of which 15% were childless and 85% fathers (table 1). The mean age in the cohort was 57 years at baseline. The baseline sociodemographic and lifestyle characteristics were

**Table 2** OR with 95% CI of MetS components in childless men (n=333) compared with fathers (n=1 817) at baseline

| | No of cases among childless men (vs fathers) | OR (95% CI) |
|---|---|---|
| Hyperglycaemia* | 88 (335) | 1.59 (1.21 to 2.08) |
| Hypo-HDL cholesterolaemia† | 102 (670) | 0.78 (0.60 to 1.01) |
| Hyperlipidaemia‡ | 97 (515) | 1.04 (0.80 to 1.34) |
| Waist circumference >102 cm | 54 (270) | 1.11 (0.81 to 1.53) |
| Hypertension§ | 222 (1160) | 1.13 (0.88 to 1.45) |

*Hyperglycaemia defined as a fasting blood glucose level ≥5.6 mmol/L or by the use of antidiabetic medicine.
†Hypo-HDL cholesterolaemia defined as HDL <1.03 mmol/L or by the use of drug treatment.
‡Hyperlipidaemia defined as triglycerides ≥1.7 mmol/L or by the use of lipid-lowering drugs.
§Hypertension (elevated blood pressure) defined by ≥130/85 mm Hg or by the use of antihypertensive drugs.
BMI, body mass index; HDL, high-density lipoprotein; MetS, metabolic syndrome.

equally distributed in general among childless men and fathers, except for the distribution of marital status, with 81% of fathers being married compared with only 37% of childless men being married.

### Association between childlessness, MetS and diabetes
#### MetS
The prevalence of MetS at baseline was 26% among childless men and 22% among fathers. The major contributing factor for MetS among childless men was hyperglycaemia (table 2). The fully adjusted analyses (model II) indicated an increased risk of MetS in childless men compared with fathers, (OR 1.22, 95% CI 0.87 to 1.72) (table 3). When comparing married childless men to married fathers, the association became stronger and statistically significant with OR 1.62 (95% CI 1.01 to 2.60) in the fully adjusted model (table 3).

#### Diabetes
The prevalence of diabetes at baseline, was 11% among childless men and 5% among fathers. In the fully adjusted analysis (model II), childless men had a higher risk of diabetes compared with fathers with OR 2.12 (95% CI 1.34 to 3.36) (table 4). The association persisted when comparing married fathers to married childless men (table 4).

### Risk of developing diabetes
The occurrence of new cases of diabetes was 20% among childless men and 22% among fathers. The mean follow-up time was 18.3 years. The fully adjusted analysis (model II) showed no increased risk of diabetes in childless men compared with fathers (table 5). However, the sensitivity analysis including only married men suggested an increased risk of diabetes among childless men

**Table 3** OR with 95% CI for MetS in childless men relative to fathers

| Total population (n=2150) | n cases | Crude OR (95% CI) | Model I* OR (95% CI) | Model II† OR (95% CI) |
|---|---|---|---|---|
| Fathers (n=1 817) | 402 (22.1%) | 1 (ref) | 1 (ref) | 1 (ref) |
| Childless men (n=333) | 85 (25.5%) | 1.21 (0.92 to 1.58) | 1.22 (0.92 to ;1.64) | 1.22 (0.87 to 1.72) |
| **Only married men (n=1515)** | n cases | Crude OR (95% CI) | Model I‡ OR (95% CI) | Model II§ OR (95% CI) |
| Fathers (n=1 392) | 317 (22.8%) | 1 (ref) | 1 (ref) | 1 (ref) |
| Childless men (n=123) | 42 (34.2%) | 1.76 (1.19 to 2.61) | 1.66 (1.11 to 2.49) | 1.62 (1.01 to 2.60) |

*Model 1: adjusted for age, marital status, SEI and education (n=1 923).
†Model 2: adjusted for age, marital status, SEI, education, BMI, alcohol (g/day), smoking and physical activity score (n=1 895).
‡Model I: adjusted for age, SEI and education (n=1 414).
§Model II: adjusted for age, SEI, education, BMI, alcohol (g/day), smoking and physical activity score (n=1 396).
BMI, body mass index; MetS, metabolic syndrome; SEI, socioeconomic index.

compared with fathers (HR 1.13, 95% CI 0.74 to 1.73) (table 5).

## DISCUSSION
### Main findings
Our study demonstrates cross-sectional associations between male childlessness, MetS and diabetes. As expected, associations were generally stronger in analyses restricted to married men where a lack of reproductive opportunities could have been accounted for. The increased risk of MetS and diabetes among childless men could not be attributed to differences in lifestyle or sociodemographic characteristics between childless men and fathers.

### Prior literature
Our findings are comparable with other cross-sectional studies reporting higher rates of medical comorbidities, poorer general health status and type 2 diabetes among infertile men.[15 28–30] However, whether infertility comes before diabetes or MetS and vice versa is still unclear. Some

studies suggest hyperglycaemia which is a central element in both diabetes and MetS to affect the endocrine control of male reproductive function, and to impair spermatogenesis, sperm maturation, erectile function and ejaculation,[11 13 31] and this makes reverse causation of our study plausible but results regarding the impact of MetS and diabetes on semen quality are conflicting.[13 32]

Furthermore, our cross-sectional findings are also consistent with reports from the USA[3] and Denmark[7] that show male factor infertility to be associated with a 30%–45% higher risk of diabetes in prospective analyses, where the chance of reverse causation is highly limited, as exposure precedes outcome. Findings of these prospective studies are also supported by a recent Danish study[33] which found higher hospitalisation rates for diabetes among men with poor semen quality and the authors of these latter mentioned studies suggested common aetiologies for male infertility, poor semen quality and diabetes. Our prospective results which showed no additional increase in risk can seem contradictory, but the mean age of the study population was more than 20 years higher

**Table 4** OR with 95% CI for diabetes in childless men relative to men with children

| Total population (n=2150) | n cases | Crude OR (95% CI) | Model I* OR (95% CI) | Model II† OR (95% CI) |
|---|---|---|---|---|
| Fathers (n=1 817) | 87 (4.8%) | 1 (ref) | 1 (ref) | 1 (ref) |
| Childless men (n=333) | 35 (10.5%) | 2.34 (1.55 to 3.52) | 2.26 (1.44 to 3.54) | 2.12 (1.34 to 3.36) |
| **Only married men (n=1515)** | n cases | Crude OR (95% CI) | Model I‡ OR (95% CI) | Model II§ OR (95% CI) |
| Fathers (n=1 392) | 64 (4.6%) | 1 (ref) | 1 (ref) | 1 (ref) |
| Childless men (n=123) | 13 (10.6%) | 2.45 (1.30 to 4.59) | 2.29 (1.18 to 4.43) | 2.05 (1.03 to 4.08) |

*Model I: adjusted for age, marital status, SEI and education (n=1 923).
†Model II: adjusted for age, marital status, SEI, education, BMI, alcohol (g/day), smoking and physical activity score (n=1 895).
‡Model I: adjusted for age, SEI and education (n=1 414).
§Model II: adjusted for age, SEI, education, BMI, alcohol (g/day), smoking and physical activity score (n=1 396).
BMI, body mass index; SEI, socioeconomic index.

**Table 5**  HR with 95% CI for diabetes in childless men relative to fathers

| Total population  (n=2028) | n cases | Crude HR (95% CI) | Model I* HR (95% CI) | Model II† HR (95% CI) |
|---|---|---|---|---|
| Fathers (n=1 730) | 373 (21.6%) | 1 (ref) | 1 (ref) | 1 (ref) |
| Childless men (n=298) | 60 (20.1%) | 1.05 (0.80 to 1.38) | 1.12 (0,85 to 1.49) | 1.02 (0.76 to 1.37) |

| Only married men  (n=1438) | n cases | Crude HR (95% CI) | Model I‡ HR (95% CI) | Model II§ HR (95% CI) |
|---|---|---|---|---|
| Fathers (n=1 328) | 281 (21.2%) | 1 (ref) | 1 (ref) | 1 (ref) |
| Childless men (n=110) | 24 (21.8%) | 1.17 (0.77 to 1.77) | 1.20 (0,79 to 1.83) | 1.13 (0.74 to 1.73) |

*Model I: adjusted for age, marital status, SEI and education (n=1 820).
†Model II: adjusted for age, marital status, SEI, education, BMI, alcohol (g/day), smoking and physical activity score (n=1 794).
‡Model I: adjusted for age, SEI and education (n=1 346).
§Model II: adjusted for age, SEI, education, BMI, alcohol (g/day), smoking and physical activity score (n=1 330).
BMI, body mass index; SEI, socioeconomic index.

than in the mentioned studies, and this makes detection of early onset of diabetes difficult.

## Mechanisms

The relationship between male reproductive health and somatic health is rather complex and different causal mechanisms and common aetiologies have been proposed for the association, namely shared genetic origins, in utero, hormonal or environmental/lifestyle factors.[1–3 7 33] More than 150 genes are linked to male infertility and simultaneously involved in pathways important for several diseases, such as cancer, cardiovascular and metabolic disorders.[34] Also, as male germ cell differentiation includes expression of up to 4% of all mammalian genes, it seems plausible that mutations in these genes could cause or contribute both to infertility, MetS and diabetes.[35] Further, as our cross-sectional results display a strong increase in risk of diabetes until the mean age of 57 years, and no additional increase in diabetes risk hereafter, as seen in our prospective results, this points to early onset of diabetes and support the hypothesis of genetic origins, as genetic predisposition increases the risk of early onset type 2 diabetes.[36] In addition to genetic defects, maternal health behaviour and environmental exposures during pregnancy have been hypothesised to act directly or through epigenetic mechanisms on the fetus and thereby influence both male reproductive health and somatic health.[2 37]

Low testosterone levels have also been associated with infertility and an increased risk of MetS and diabetes, as testosterone plays an import role in glucose and lipid metabolism.[38–40] One study found low testosterone to predict development of MetS and onset of diabetes, even after adjusting for BMI, insulin resistance and other established risk factors for these conditions.[39 41] Another study among men with infertility problems and decreased sperm counts, reported low testosterone levels to be associated with higher levels of HbA$_{1c}$.[42] However, obesity, which is a strong risk factor for MetS and diabetes, is also known to lower testosterone levels, as testosterone is converted to oestradiol in adipose tissue. On the other

hand, low testosterone levels are also known to increase obesity.[43] The causality of the relationship between low testosterone, MetS and diabetes is therefore unclear but may be bidirectional.[38]

Low socioeconomic status and adverse lifestyle factors have been suspected to explain the association between poor reproductive health and somatic health. In the present study, we adjusted for multiple lifestyle factors and socioeconomic status, and associations were still seen. However, the operationalisation of lifestyle factors may not sufficiently reduce the risk of confounding in the analysis of diabetes risk among childless men. This is due to the fact that information on lifestyle factors were only obtained at baseline and men with a diabetes diagnosis prior to baseline could have changed their lifestyle according to the disease. This makes it impossible to distinguish between prediagnostic lifestyle and postdiagnostic lifestyle, and only the lifestyle related to both exposure and outcome, namely the prediagnostic lifestyle, qualifies as a confounder. This may have induced some residual confounding in the analysis of diabetes risk, but not in the analysis of MetS risk as the diagnosis was based on information collected at baseline. Furthermore, our prospective analyses, which did not display the same association as our cross-sectional analyses, could also be influenced by results of clinical examination at baseline, and a wish among childless men to live a healthier life style. Data regarding MetS criteria were only available from the baseline clinical examination, why a prospective analysis of the association between childlessness and MetS could not be performed.

## Strengths and limitations

The present study has several strengths. First, the MDC-CC is sampled from the background urban population, meaning that men from all socioeconomic backgrounds were represented. A previous study showed no difference in baseline sociodemographic characteristics compared with study participants in the MDC.[21] Also, among Swedish men born between 1935 and 1945, one in six remained childless[44] which is comparable with the

proportion of childless men in our cohort. Second, the mean cohort age at baseline was already advanced, and childlessness at an advanced age strengthens the assumption of infertility. Third, the valid and comprehensive national Swedish registers provided information on emigrations, death and disease-limited loss to follow-up and made long-term follow-up of more than 18 years in average possible—the longest mean follow-up among the few similar studies published to date.[3 7 33]

Our study also has limitations. As for cross-sectional studies in general, data regarding exposure and outcome were collected simultaneously, making it impossible to rule out reverse causation, However, as previously mentioned, studies have reported prospective associations between male infertility and risk of diabetes, in younger study populations than the present cohort,[3 7 33] and the chance of reverse causation in prospective studies is low.

Furthermore, using childlessness as a proxy of infertility poses some challenges. The group of childless men is heterogeneous in relation to different causes of childlessness and do not only reflect male infertility (eg, voluntarily childlessness, homosexual men, men with infertile partners or single men). As only one in four childless Swedish men reported childlessness to be volitional,[44] this limits the risk of our sample to reflect voluntary childlessness.

Also, we were unable to distinguish between fathers of biological or adopted children and fathers of children conceived after in vitro fertilisation (IVF). However, as adoption rates in Sweden were rather low at the time of baseline,[45] misclassification would attenuate the estimate towards the null. Likewise, the chance of having children conceived after IVF was insignificant as assisted reproduction as treatment of impaired male fertility was not widely performed before and at the time of baseline in 1991–1994.[46] This strengthens the usefulness of childlessness as a proxy for infertility in the present cohort.

Misclassifications can bias the estimated associations and the question is, whether men with MetS and diabetes are systematically more likely to be classified as childless. For instance if men with low socioeconomic status are systematically more likely childless due to being single, and at higher risk of disease, this could lead to an overestimation of the association between infertility and MetS/diabetes. However, by adjusting for socioeconomic status, and associated lifestyle factors as well as by confirming our findings in sensitivity analyses based on married men only, this source of bias was minimised.

Unfortunately, comparison of male participants versus male non-participants in either the MDC or the MDC-CC have not been done, but a study from 2001 concluded that mortality for men and women combined was higher in non-participants than in participants which could reflect healthy selection bias in the cohort.[21] Selection bias occurs when conditioning on a common effect of exposure and outcome—in this case conditioning on survival of participants. In our analyses, selection bias could occur if fatherhood status and MetS or diabetes are directly or indirectly related to preterm death before initial enrolment into the cohort. For example, if childless men are more likely to have lower socioeconomic status (which itself is associated with increased risk of preterm death), and if men with MetS or diabetes are more likely to have comorbidities (which is related to preterm death).

## CONCLUSION

In conclusion, our study showed a higher risk of MetS and diabetes among childless middle-aged men that could not be explained by differences in lifestyle, sociodemographic characteristics or health-seeking behaviour. This may support the hypothesis that a man's reproductive health is closely intertwined with his somatic health, however due to the nature of the cross-sectional design, where information on exposure and outcome is collected simultaneously, reverse causation cannot be excluded. While using childlessness as a proxy of male infertility may cause misclassification bias, it may still provide insight into a man's risk of disease. The simple objective measure of exposure enables for future studies to examine the association between male reproductive health and somatic health in large population-based cohorts.

**Contributors** JPB and AG acquired funding for the study. ABB, CHG, SST, AG and JPB designed the study. ABB, SST and CHG analysed data and ABB wrote the manuscript. PMN contributed with acquisition of data, critical discussion and revision of the paper. All authors contributed to data analysis/interpretation, critical revision of the paper and final approval of the manuscript.

**Funding** This study was funded by ReproUnion and also supported by the Medical Research Council of Sweden (grant K2011-65X-20752-04-6), the Region Skåne County Council, the Ernhold Lundstrom Foundation, for the MDC-CC follow-up clinical examination.

**Competing interests** None declared.

**Patient consent** Not required.

**Ethics approval** Ethics Committee of the Lund University (LU 51-90) and the Swedish Data Inspection Agency.

**Provenance and peer review** Not commissioned; externally peer reviewed.

**Data sharing statement** No additional unpublished data.

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
