## [Reviewer comments · BMJ Open]

ARTICLE DETAILS

TITLE (PROVISIONAL)	Risk of metabolic disorders in childless men - A population-based cohort study
AUTHORS	Bungum, Ane; Glazer, Clara; Bonde, Jens Peter; Nilsson, Peter; Giwercman, Aleksander; Søgaard Tøttenborg, Sandra

VERSION 1 – REVIEW

REVIEWER	Tina Kold Jensen University of Southern Denmark, Denmark
REVIEW RETURNED	06-Mar-2018

GENERAL COMMENTS	In the article the association between childlessness and metabolic syndrome markers and DM is studied cross-sectionally and prospectively in a cohort of middle aged men from Malmö. Childless men had a higher prevalence of hyperglycaemia. In addition, childless men had a higher prevalence of DM and childless and married men had a higher prevalence of Mets. In the prospective follow-up no association with DM was found in childless men. The study is well performed and the findings interesting and of public health relevance. I, however, have some concerns which need addressing before I can recommend publication. My main concern is the conclusion of the study. Generally, the cross-sectional findings are difficult to interpret and even though the authors state that the adjust for confounders and findings cannot be attributed to these, as information on outcome and exposure is collected simultaneously, reverse causation cannot be excluded. As no association is found in the strongest design, the follow-up, I am not convinced that the conclusion should be as strong. Generally, men with Mets factors and DM at the inclusion may have an unhealthier lifestyle or may even have changed their lifestyle according to the disease. In addition, Mets factors or DM may have caused the man to be childless thereby being a cause of and not a consequence of the infertility. Generally, the description of the cohort may be expanded. Selection due to preterm death before inclusion and during follow-up may be a cause of selection that needs description. Likewise, not all the men may have finished their reproduction and it is difficult to understand who men how are childless at the start but who become fathers during follow-up are treated. Some men may have Mets risk factors at the start and then develop DM during follow-up, how are they treated? Normally Mets risk factors precede DM and men with one or more risk factor should be treated differently to men with no Mets factors as inclusion (or at
---

	least Mets should be adjusted for). The DAGS diagram would be interesting to see. On page 11 the authors state that the incidence of DM is respectively 20% and 22%, incidence is not percent rather new cases/risk time. The authors argue that the use of childlessness as a marker of infertility may cause non-differential misclassification as it is not influenced by the disease, I however, believe that the diseases may cause childlessness either due to biological or social factors (they are more often unskilled workers) thereby causing differential misclassification. Minor point: The mean length of follow-up should be stated in the abstract. In table II cases/controls could be seen in one column and OR presented leaving out columns of 1 (ref).
--	--

VERSION 1 – AUTHOR RESPONSE

Reviewer: 1

1. My main concern is the conclusion of the study. Generally, the cross-sectional findings are difficult to interpret and even though the authors state that the adjust for confounders and findings cannot be attributed to these, as information on outcome and exposure is collected simultaneously, reverse causation cannot be excluded. As no association is found in the strongest design, the follow-up, I am not convinced that the conclusion should be as strong.

Authors response:

We agree that reverse causation cannot be excluded due to the cross-sectional design, and therefore been stated in the conclusion and elaborated in the discussion of the revised version.

2. Generally, men with Mets factors and DM at the inclusion may have an unhealthier lifestyle or may even have changed their lifestyle according to the disease. In addition, Mets factors or DM may have caused the man to be childless thereby being a cause of and not a consequence of the infertility.

Authors response:

We agree that the operationalization of lifestyle may not sufficiently reduce the risk of confounding. Given that data on lifestyle factors are only obtained at baseline we are unable to distinguish between pre-diagnostic lifestyle (i.e. the lifestyle related to both exposure and outcome and consequently qualifies as a confounder) and post-diagnostic lifestyle which in the cross-sectional set-up can only be related to the exposure and thus not qualify as a

confounder which would result in residual confounding of the results. However, we believe the bias due to confounding in the cross-sectional analysis is limited to men with a diabetes diagnoses at inclusion, as the MetS diagnosis is based on clinical information collected at baseline giving the men no chance to change their lifestyle according to MetS, i.e. they all have pre-diagnostic lifestyle.

In regards to the prospective analysis with diabetes as an outcome, men who received a diagnosis of MetS based on the clinical examination at inclusion could subsequently change their lifestyle, and thereby reduce their risk of diabetes, and with the lack of several data points on lifestyle and consequently no way to treat lifestyle as a time dependent covariate, the risk of residual confounding remains. We have elaborated on this limitation in the discussion.

3. Generally, the description of the cohort may be expanded. Selection due to preterm death before inclusion and during follow-up may be a cause of selection that needs description. Likewise, not all the men may have finished their reproduction and it is difficult to understand who men how are childless at the start but who become fathers during follow-up are treated.

Authors response:

In the revised version of the manuscript we have updated Figure 1, so the figure only describes the recruitment of men to MDC and MDC-CC.

Also, we agree that potential selection bias require more elaboration. Unfortunately there has been no comparison of male participants versus male non-participants in either MDC or the MDC-CC, but a study from 2001 concluded that mortality for men and women combined was higher in non-participants than in participants during both recruitment and follow-up. Selection bias occurs when conditioning on a common effect of exposure and outcome. In this specific case selection bias could occur if father status and MetS/diabetes is directly or indirectly related to preterm death (either before initial enrollment into MDC or before subsequent enrollment into MDC-CC). E.g. if childless men are more likely to have lower socioeconomic status (SES) which itself is related to preterm death, and if men with MetS/Diabetes are more likely to have comorbidities which affects the risk of preterm death. This would cause selection bias creating a spurious association between childlessness and Mets/Diabetes.

Regarding men who were childless at the study entry but who become fathers during follow-up are treated, we had stated in the first version of the manuscript that two men became fathers after baseline and thus treated as 'Childless' on page 11.

4. Some men may have Mets risk factors at the start and then develop DM during follow-up, how are

they treated? Normally Mets risk factors proceed DM and men with one or more risk factor should be treated differently to men with no Mets factors as inclusion (or at least Mets should be adjusted for). The DAGS diagram would be interesting to see.

Authors response:

We agree that MetS and diabetes are disorders that are highly correlated, and may even coexist, and therefore we believe adjusting for MetS would cause over-adjustment. Please see DAG below.

5. On page 11 the authors state that the incidence of DM is respectively 20% and 22%, incidence is not percent rather new cases/risk time.

Authors response:

This has been corrected to “the occurrence of new cases of diabetes was 20% among childless men and 22% among fathers”.

6. The authors argue that the use of childlessness as a marker of infertility may cause non-differential misclassification as it is not influenced by the disease, I however, believe that the diseases may cause childlessness either due to biological or social factors (they are more often unskilled workers) thereby causing differential misclassification.

Authors response:

If the reviewer is correct and the diseases in fact cause infertility this would be a matter of reverse causation rather than misclassification – something inherently difficult to rule out in a cross-sectional analysis as previously discussed. However, the reviewer also touches upon, infertility and MetS/diabetes, may be caused by shared underlying biological factors, which we have not been able to adjust for in the present study. In regards to misclassification, the question is, whether men with MetS and diabetes are systematically more likely to be classified as childless? If unskilled workers/ low SES men are systematically more likely to be deselected by women (thus childless due to being partnerless rather than truly infertile), and at higher risk of disease because of lifestyle, this could lead to an overestimation of the association between infertility and MetS/Diabetes. By adjusting the analyses for socioeconomic status and associated lifestyle factors this source of bias was minimized. Further, the strengthened associations in our sensitivity analyses comparing childless married men with married fathers suggest that the observed associations are not merely a function of low SES and implicit “single” life.

7. The mean length of follow-up should be stated in the abstract.

Authors response:

The mean length of follow-up is now stated in the abstract.

8. In table II cases/controls could be seen in one column and OR presented leaving out columns of 1 (ref).

Authors response:

Table II have been updated according to this advice.

VERSION 2 – REVIEW

REVIEWER	Tina Kold Jensen University of Southern Denmark, Denmark
REVIEW RETURNED	03-Jul-2018
GENERAL COMMENTS	The authors have adequately addressed my comments